# Basic Hazard Control Plan for Small Wild Ungulates Slaughtered for Meat Production

**DOI:** 10.3390/foods12071511

**Published:** 2023-04-03

**Authors:** Davies Veli Nkosi, Johan Leon Bekker, Louwrens Christiaan Hoffman

**Affiliations:** 1Department of Environmental Health, Tshwane University of Technology, Pretoria 0183, South Africa; 2Department of Animal Sciences, University of Stellenbosch, Matieland, Stellenbosch 7602, South Africa; 3Centre for Nutrition and Food Sciences, Queensland Alliance for Agriculture and Food Innovation (QAAFI), The University of Queensland, Digital Agricultural Building 8115, Office 110, Gatton 4343, Australia

**Keywords:** hazards, slaughter, hazards microbial, chemical, physical, small wild ungulates, consumer

## Abstract

Animal slaughter plans and related activities must not increase the number of hazards in meat. In their nature, these processes must reduce possible hazards to minimum or acceptable levels. This is a generally accepted concept worldwide; hence, authorities continue to develop regulations that seek to mitigate the scourge of meat hazards for consumer protection. The situation is similar with small wild ungulate meat, in which a hazard analysis plan needs developing to improve meat safety. This investigation follows a narrative review of articles published for a PhD program and other scholarly articles supporting the concept of a basic slaughter plan for small wild ungulate animals in South Africa. The findings of this investigation highlight the need to control hazards within one health concept plan, which should be implemented and propagated by establishing forums that will drive meat safety solutions in these communities. There should be a basic hygiene slaughter plan developed and endorsed by all members of the forum. The outcome must be the control of microbiological, chemical and physical hazards from farm-to-fork, and as part of a system imbedded in game meat policies and regulations.

## 1. Introduction

Globally, the game industry has evolved into economically strong ecotourism, hunting, breeding and game meat production industry, its supply to the local and international markets is growing [1]. This growth is supported by the belief that the wildlife industry could contribute significantly to food security and provide a healthy source of protein to consumers [2]. Researchers, such as Taylor et al., reported that in South Africa, out of all game animals hunted for meat production between 2013 and 2015, approximately 41% were impalas, a species that is part of the ‘small wild game meat animals’ category [3]. This situation is not different in many game meat trading countries, where small wild ungulates are kept in natural veld and are free-roaming in small to larger predator-controlled camps or farms [4]. Controlled boundaries/fences are in place to ensure the continued production of the targeted species, for the monitoring of grazing, control of hunting, facilitation of breeding and self-sustainability of the game population in the farms [3]. However, surplus animals are generally hunted as part of the game management plan by trophy hunters, or culled for meat production by private hunters or commercial game meat organizations [5]. During these periods of killing, the two most common methods employed include the use of a single projectile, such as a 30.06 rifle (150 g Nosler Accubond Points), targeting the head, neck or chest cavity and a shotgun utilizing numerous pellet sizes, such as a 12-gauge shotgun that utilizes 35 g cartridges with a diameter of 5.2 mm per pellet, firing 44 pellets per round from a helicopter [6]. In formal processes of hunting or harvesting for commercial purposes, the game animal is killed, transported to a facility or registered game abattoir for secondary dressing and processing, such as the removal of the hide, evisceration (if not done in the veld after killing, where basic biosecurity measures, such as the use of different knives, sterilization of knives and ensuring that evisceration cuts are performed so that no contamination is transferable to muscles of carcasses during cutting, would have taken place to reduce the risks of contamination and spread of diseases), meat inspection and chilling [7]. These processes differ from the killing of game by farmers for private usage, where compliance to specific rules and regulations would differ [8,9]. These processes of killing in the wild to chilling after dressing at an abattoir may lead to the contamination of meat by different hazards, such as microorganisms, chemicals from ammunition and physical bullets fragmenting and dispersion [10]; this contamination could be ingested by an unsuspecting consumer, thus leading to other health impacts [6,11]. The hazards contributed by the two killing methods generally used, single-projectile targeting the thorax (for ordinary hunters) and aerial (helicopter) shotgun targeting the head (for the commercial harvesting or culling of animal surplus or animals not deemed suitable for hunting), have been investigated [6,11,12,13,14].

In the past, there has been no integrated package that outlines the processes of hazard identification, evaluation and control in small wild ungulates slaughtered for private or commercial purposes, especially in developing countries [15,16]. This paper consolidates the results of three investigations of impala (*Aepyceros melampus*) during a commercial slaughter program at a game farm in South Africa, and provides a summary of the hazards and control plans adoptable for the slaughter of small game meat animals. These investigations document the presence of microbiological organisms (*Total Plate Count*, *coliforms*, *Escherichia coli*, *Salmonella*), and the role that organic acids (lactic LA and acetic AA) could have on their reduction on fresh carcass wounds when applied during slaughter [6], highlight and quantify toxic metals (lead, mercury, cadmium and arsenic) introduced by the different bullets and ammunition types during the killing process [12], and measure physical hazards (bullet particles and bone fragments remaining after meat inspection) in impala carcasses killed by either rifle or shotgun (thoracic or head) shots fired from a helicopter [14,17].

To develop a comprehensive game meat safety plan, different stakeholders within the animal meat production cycle need identifying, and from these stakeholders, there must be specific responsibilities undertaken to promote meat safety awareness from farm-to-fork, especially in countries where the consumption of game meat is common [18,19]. Information, such as a that about basic hazard control, must be available and should be supported by scientific information or knowledge [20,21].

## 2. General Food Safety Requirements

In most instances, countries have specific requirements related to the primary production of food, such as good hygiene practices (GHP) and the use of a food safety management system (FSMS), such as hazard analysis critical control plans (HACCP) and ISO 22000:2018. Furthermore, the Codex Alimetarius provides general principles of food hygiene [22] and a code of hygienic practice for meat production [23], and the European Union legislation provides rules for food business operators on the hygiene of foodstuffs [24], rules on the hygiene of food of animal origins [25], and controls on products of animal origin intended for human consumption [26]. Standards, such as ISO 22000:2018, set pre-required plans to be addressed by food processing establishments, to ensure the production of safer products [20]. These requirements are clear on the expectations for food producing facilities. Evidently, food-producing facilities have to document a FSMS comprising policies and procedures for hazard identification, evaluation and control. This system must detail mitigation measures or the monitoring of adoptable plans for the effective reduction of hazards to acceptable levels or their total elimination before the product reaches the consumer [27]. Within the context of this review, microbiological hazards, for example, relate to zoonotic diseases and other pathogens, such as brucellosis and salmonella [6,11], physical hazards relate to bullet particles and bone splinters resulting from shooting [14,17], and toxic metals [12,13] in meat relate to those from the environment and to the contamination caused by the ammunition selected for killing, and are left in carcasses during slaughter [28,29,30].

## 3. Small Wild Ungulate Killing Environments

Hazards found in small wild ungulate carcasses could emanate from three environments that are addressed in the one health concept [31]. Figure 1 illustrates the roles of the different areas of control required to achieve the one health objectives: (1) Farm health—this is the intensive or extensive game farm environment where the hunting or harvesting of game animals takes place. (2) Animal health practices—in killing, slaughter and dressing. (3) Human health—of slaughter operators and consumers [32]. In essence, the idea of one health is centered around two important pillars, namely (1) achieving maximum health outcomes for all role players and stakeholders in game meat production, and (2) the recognition of the interconnection between people, animals/plants, and their shared environment [33].

## 4. Material and Methodology

This paper adopted a review consolidated with scientific evidence from research conducted on hunted or harvested small game wild ungulates such as impala (*Aepyceros melampus*).

Different conclusions from the main project “Strategies to reduce microbial and physical hazards on commercially harvested impala (*Aepyceros melampus*) carcasses” are drawn in and packaged to develop this generic small wild ungulate meat animal slaughter plan that could be adoptable for the same category sizes (‘small game’) by game abattoirs, slaughter operators, hunters and the hunting community, the government and other role players from farm-to-fork. While this study focused on information gathered from published articles, as part of a PhD program in Environmental Health from 2020–2022 and approved by the Animal Research Ethics Committee of Tshwane University of Technology (Proposal code AREC 2019/03/002), there were other relevant scholarly articles included to support the presented information. The approach was to search for articles that could contribute to the development of a basic wild game animal killing plan for the purpose of meat production. Expertise in game meat slaughter, published work, the guidelines on the process flow of game slaughter from farm to abattoir, and possible intervention strategies implementable to ensure or to facilitate cleaner small wild ungulate meat production processes were the sources for guidance for these suggestions.

## 5. Results

The results of this study encompass two important components of safety provision for game meat. These include the theoretical model for the slaughter of game meat animals, and the basic meat hygiene plan for the slaughter of small game animals. The development of this slaughter plan was carried out in a way that any abattoir slaughtering the same category of species could adopt and effectively use the plan.

## 6. Theoretical Model for Game Animal’s Slaughter and Meat Safety Assurance

Figure 2 illustrates a theory model for the slaughter of small wild ungulate animals, such as impala, hunted or culled/harvested for meat production in South Africa. This model identifies stakeholders and role players in the game meat production industry, with the aim of ensuring an integrated approach to hazard identification and possible hazard control by elimination or reduction to required levels across all small wild ungulates killed for meat production [14]. Researchers [32,33] identified the need for the establishment of working committees that collaborate with stakeholders and identified role players with the fundamental role of developing responses for the control of hazards during game meat production processes. Figure 2 depicts three pillars of the game meat assurance plan: (1) inputters—these are phenomena this forum will discuss in order to influence good policies for the control of game meat slaughter and subsequent hazards; (2) forum members—these include the stakeholders and role players in processes of game meat production, and the expectation is that with correctly constituted and frequent collaboration by industry stakeholders and the forum itself, policy directions to improve the identification, evaluation and control of hazards found during slaughter could be improved; (3) outputs—these are expected outcomes or policies implementable by the committee and all stakeholders. It is important that the industry is the driver of these forums to ensure it continues to address issues of meat safety and any other related responsibilities, such as training, capacity building, etc. The role of the government should be to facilitate and ensure the efficiencies of outputs.

Table 1 clarifies the generic hazard identification and control plan for the slaughter of small wild ungulate meat animals from killing at a farm to after chilling at an abattoir. The abovementioned stakeholders in the game meat industry (Figure 2) can use this hygiene control/hazard control plan for all meat production plans. The outcomes of the critical analysis can function as a guide for further deliberations by private hunters, commercial game meat producers and meat producers from trophy hunting. The effective implementation of the plan will assist with adding knowledge and information on the production of safer meat to the game meat industry, thus ensuring that meat produced from these processes is safer for human consumption. This plan identifies different hurdles to the control of biological, chemical (toxic metal) and physical bullet fragments and bone splinters, which this hazard control plan can control.

## 7. Discussion

Meat produced from small game slaughter could be categorized in three segments: (1) meat for private usage by farmers, (2) meat that is commercially harvested and (3) meat derived from trophy hunting [35]. The suggested model for small game slaughter and meat safety assurances could be adoptable in all three segments of game meat production [36]. As is typical in processes involving many outcomes, an integrated approach to pave the way for the common awareness and understanding of different roles is drawn from the different stakeholders involved in game meat production [37,38]. To support their comradeship, one suggestion is the model of stakeholders in a forum. This forum will be tasked with the promotion of the one health concept, as outlined by the World Health Organization (WHO), when dealing with ensuring the linkages of the three environments to facilitate consumer health and hygiene compliances [32,33]. This work includes the control of hazards found in the three environments of game meat animal production, the animal’s health, environmental health and human health, involved in all processes from farm-to-fork to ensure the production of hygiene-complying game meat. While scientific papers present the generally acceptable logic of this control, one must not forget the existence of a non-exhaustive list of materials that stakeholders could use to improve the provision of safer meat products. These may include:Published and non-published scientific material referenced in different theses, and other relevant scholarly search engines,National and international standards/industry standards,A framework and guidelines, such as the five keys to safer food production.

## 8. The Hygiene of the Environment

In wildlife farms, game breeding for sport or trophy hunting, and subsequently game meat production, occurs as part of a game farm’s resource management practices. Meat derived from these processes ends up being consumed by the formal or the informal markets. Prior to these activities, however, several environmental issues must be addressed to ensure the killing of desired animals and that the farms and the farming environments are fit to keep game animals [39]. In the context of this study, these could include issues such as farms with no controlled diseases, safe water, feed and land uncontaminated by previous mining and anthropogenic activities [40]. These processes need monitoring to ensure there are no hazards introduced to meat animals during their rearing on the farm level.

## 9. Animal Health

A number of important animal health practices and principles must be applied during meat animal management, and may include the access to veterinary care and subsequent medicine control [31]. Animal farming plays an important role in the appearance and the spread of zoonotic diseases, as numerous common infectious diseases reach humans. Similarly, wild animals are an important reservoir of infectious diseases, and most of the zoonotic pathogens originate from wildlife [41]. Emphasis must be placed on improved processes of maintaining animal health to reduce hazards introducible during animal handling and killing, and subsequent meat production. The promotion of good and traceable meat products from farm to fork can be advantageous to the industry, as various studies have shown that consumers are becoming more conscious about what they eat and are willing to pay a higher price for meat with information about its farming and slaughter practices [42,43]. However, it is challenging to link broader health issues to a specific animal due to the nature of these animals—all being wild. Thus, this plan should encompass not only an individual animal’s health status, but also that of all the animals (irrespective of species) on that property. Obviously, a well-trained meat inspector/veterinarian inspecting each carcass and the relevant offal will aid in the identification of animals that do not meet the health requirements. The challenge is that in many developing countries, there are not enough trained staff to undertake these inspections, whilst even more scarce are laboratories to help with the diagnosis of possible hazards.

## 10. Human Health Assurance

Several factors need addressing to ensure the safeguarding of consumers. This includes the hygiene of the environment (abattoir), including that of people killing and dressing the animal, meat inspectors and consumers as final end users [43,44,45,46]. The identification of meat safety threats, the implementation of control measures and where necessary, the development of game meat legislation, are important in the abattoir industry [41]. In the game meat context, meat safety assurance relates to the detection and identification of hazards during in-field killing, that are likely to be present in the slaughter processes and could harm the people handling the animal during slaughter operations [47]. Evidently, this assurance should be stretched to consumers and their practices during final handling and preparation at home [48]. It is through addressing these processes that game meat safety could be achieved. Researchers [19,28,49] have continuously reported on the involvement of other stakeholders in the promotion of safer game meat. This investigation suggests that the stakeholders listed below are important for small wild ungulate meat production, and could play a significant role in the formation of a stakeholders’ forum, as outlined in Figure 2.

## 11. Regulatory Authorities

The role of regulatory authorities in outlining the requirements for safe game meat production remains important [50]. This responsibility is inherent to all spheres of the meat production process. Regulatory authorities must ensure that processes of meat production are regulated; these include the production of healthy animals (animal health) and safe processes of meat production (veterinary public heath) [51]. The killing of animals must be carried out in a humane manner by proficient hunters, and for meat production, such animals must be dressed at a legally registered facility where there can be inspected by trained meat inspectors. Even after proper slaughter, the authority of controlling the distribution and sale of the meat is the responsibility of municipal health services (environmental health practitioners), where the registration of processing establishments is necessary [52]. Regulatory authorities, in conjunction with the game meat industry, are therefore compelled to establish regulatory standards to minimize the risks associated with the production of game meat, which must also adhere to the food safety requirements of the legislation and standards above. The basis for legislation and standards should be on sound scientific processes that place the concern about food contamination into proper perspective, and thereby ensure equitable enforcement. In other countries, such as South Africa, “assignees”, which are companies or organizations tasked with conducting specific services to ensure or promote meat safety production [53], have the responsibility of ensuring meat inspection for specific meat species. The responsibility to monitor the performance of these assigned organizations lies within veterinary public health, and the participation of both organizations as stakeholders is important [54].

## 12. National Parks and Game Conservation Environments

National parks generally belong to government/s, and are administered by boards mandated to maintain wilderness areas and public nature reserves [55]. In addition, several game farmers play an important role in the control and monitoring of wildlife animals in their private capacity. These private game farms belong to private owners and are managed in line with the existing regulations of the country [56]. In general, both national parks and game farmers will allow the harvesting of game as part of their game management plans [57]. Normally, the meat of animals killed in national parks, as part of management strategies, goes to the park and surrounding communities as part of their endeavor to enhance protein supply to needy communities. The situation is different with private game farms, where meat derived from controlled hunting (including trophy hunting) and harvesting is distributed into the local and export markets [56]. Stakeholders from the two game-keeping establishments should play a significant role in environmental conservation management and the production of meat safe for human consumption.

## 13. Game Farmers and Hunters’ Associations

Game animals destined for meat production through trophy hunting and traditional hunting plans are generally killed by professional or traditional hunters, and their roles in the processes of hunting could have meaningful contributions to the developed model [56]. This review has shown that there can be a significant number of hazards introduced by the killing methods adopted. Hunters, in collaboration with wildlife farmers, need to ensure that the killing of animals destined for meat production is controlled, and the slaughtering and processing of such meat is carried out in a hygienic manner. As with most industries, there are a number of associations that play a role in wild-life breeding and the sustainable production of wild animals [58]; these, for example, include game farmers’ associations, hunters’ associations, game breeders, etc. Generally, the roles of these associations are to promote the sustainable usage of game and of the game meat animal ecosystem, production of game trophies, and sustainable eco-tourism. According to Brown [59], associations and their constituent networks can act as “bridging organizations” in an emerging multispectral paradigm by creating horizontal connections, increasing grassroots influence on policy, and disseminating new information.

## 14. Stakeholders in the Game Meat Trade

Trading with wild meat is a new and enticing business for meat producers. Generally, this is because of the view that “game meat” is predominantly a cheaper and much healthier/leaner type of meat. Retailers and processors have ‘seized the moment’ and are continuously developing different meat products from game meat [58]. Processors play a significant role in the sourcing of products or ingredients that are certified safe for human consumption [60]; these include products such as carcasses sourced from legally registered abattoirs, that have undergone hygienic slaughter, and meat inspection, and are certified safe for consumption by humans [61].

## 15. Consumers

Consumers are continuously searching for an alternative source of protein from domestic animals, and it is for this reason that the interest in and search for these must be upheld [44]. These consumers could play a significant role in shaping the future of game meat production. Their contributions include, but are not limited to, the quality traits of meat, such as the appearance, texture, taste and color of game meat at the point of sale [45], and their need for buying safe meat from a reliable source [62]. Game meat consumers are a big driver of game meat usage and for this reason, any success in game meat production is directly dependent on the acceptability and usability of game meat products [43]. It is on this level that more awareness could be shown by these consumer-based organizations and community leaders who are significant to the future of game meat production [62].

## 16. Basic Meat Safety Plan for Small Game Slaughter

The different studies, from which inferences were drawn, were on impala (*Aepyceros melampus*) species, which fall in the category of ‘small’ species, such as impala, springbok and same-sized game meat animals, as classified by the Veterinary Procedure Notice and other researchers [6,61]. The implementation of these recommendations can be on all small wild ungulate meat animals hunted, harvested or culled for meat purposes, given their similarity, unvarying size, body weight, hide thickness and their similar slaughter processes. The basic hygiene plan looks at different steps of slaughter using two types of ammunition: a single projectile and a shotgun utilizing numerous pellets for hunting or harvesting purposes. It is important to note that, these guns could differ to other guns or ammunition used for the same purposes [62]. The presented summary of the basic hygiene requirements (Table 1) provides a generic approach and guidance to where a specific hazard is likely to be found during slaughter, the type of hazards likely to be found, the source of the hazard and the possible controls employable during the slaughter of similar sized animals using the same or a similar kind of rifle and ammunition [62,63,64,65]. Additional studies would be necessary for larger game animals with thicker hides and more muscles, as these animals differ in size, hide thickness, etc., differences that create their own unique set of hazards [39,63]. Although they are not expected to be different, the ability of OA in reducing the microbial load on bullet wounds of carcasses could depend on the type of OA used as an intervention plan and their application during slaughter, which may vary from abattoir to abattoir [45].

## 17. Future Research

The study identified, but did not discuss in depth, the following areas for future research:Investigation of the use of other organic acid treatments in game meat slaughter for the purpose of microbiological decontamination of carcasses at an abattoir;Economic feasibility studies to investigate the implementation of different decontamination plans for small-scale game meat abattoirs;Mandated authorities for the control and monitoring of game meat animal slaughter, meat handling and distribution in South Africa;Development of a rapid test kit for toxic metal surveillance on carcasses at the abattoir;Investigation of the actual risk associated with the illegal slaughter of game meat and the development of strategies to narrow the gap between illegal (informal) and formal slaughter, which is generally the export market;The use of copper-made bullets as a replacement for lead-based bullets in South Africa.

## 18. Conclusions

Hunting and culling will continue to grow as the number of game conservation centers grow. This growth is directly caused by the increasing demand for game meat and availability of game meat species, which require killing as part of formalized game management plans. However, the supply chain of game meat is often complicated, as there are different stakeholders with different objectives found during the hunting season; these include hunting or culling for meat production and hunting for trophies. In all these processes, the (by)product is meat that ends up on the plates of consumers. While it could be said that meat produced from harvested or culled animals for meat purposes generally goes through stringent processes of hygiene compliance, the same could not be said about trophy-killed animals where the production of meat is of secondary importance (as obtaining a good trophy is the major focus of the hunt). The use of this suggested basic hygiene plan will ensure that stakeholders in game meat production are at least provided with an easy-to-follow plan that highlights the production of cleaner meat during the killing processes of small game.

## Figures and Tables

**Figure 1 foods-12-01511-f001:**
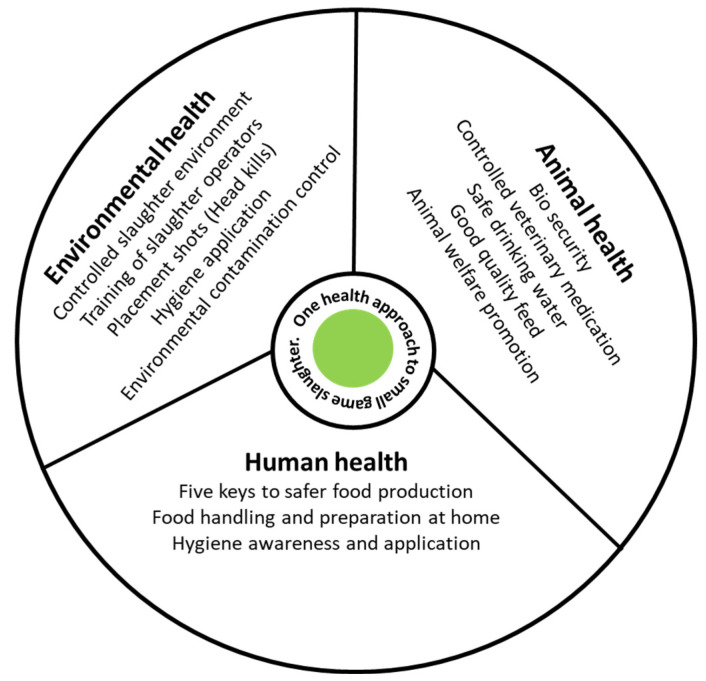
One health approach for safer small wild ungulates meat production [32,34].

**Figure 2 foods-12-01511-f002:**
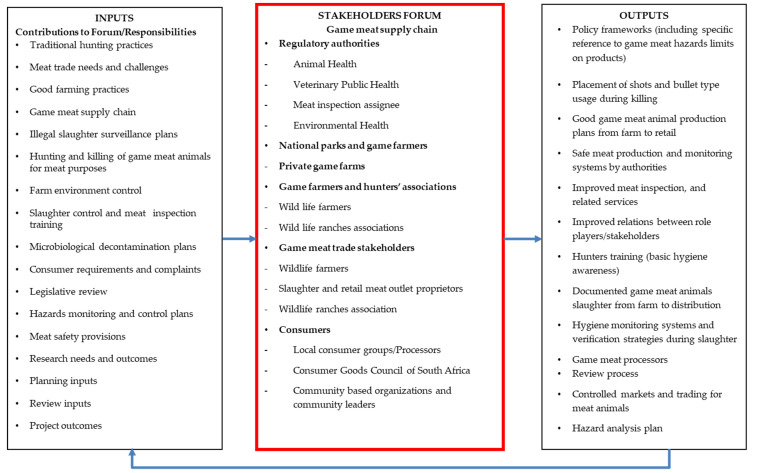
Theoretical model for small wild ungulate meat animal slaughter in South Africa.

**Table 1 foods-12-01511-t001:** Basic hazard identification and control plan for the slaughter of small wild ungulate animals from killing at a farm to after chilling at an abattoir.

Step	Description	Possible Hazards	Source of Hazard	Control and Monitoring	Reference
- Killing or hunting of game meat animals	- Game killing, hunting, harvesting or cropping	- Biological: zoonotic diseases - and pathogens	- Introduced by diseased animals and killing methods	- Monitoring of animals for disease (animal health plans) - Selection of appropriate calibers (considering animal size) to keep mutilation and subsequent contamination to the minimum - Shot placement (head versus thoracic shots) - Training hunters	[5,10]
		- Chemical: toxic metals	- Introduced through the selection and utilization of specific bullets	- Selection of bullets not composed by toxic metals, such as lead, copper, etc.	[12,13]
		- Physical: bullet particles (introduced by killing plans) and bone splinters (caused by bullets impacts)	- Introduced through the selection and utilization of specific bullets	- Selection of bullets that do not fragment - Selection of appropriate calibers (considering animal size) to minimize fragmentation of bullets and bones - Informed shooting to ensure that it is understood that for meat purposes only head shots are acceptable	[14,17]
- Bleeding/evisceration	- Game meat animals, bleeding and evisceration	- Biological: pathogens such as zoonotic ones	- Introduced through poor bleeding and poor hygiene	- Application of GHP—bleeding processes, and sterilization of bleeding knifes - Use of two-way knife system, in which one knife is used for each carcass, then washed and sterilized	[5,10]
		- Chemical	- Chemicals or oil leaking from vehicle/s	- Maintenance of vehicle and surfaces where carcasses are loaded - Use of designated transportation for hunting and loading of killed game animals	[12,13]
		- Physical	- None	- None	
- Transportation to further dressing	- Transportation to further carcass dressing at an abattoir	- Biological: pathogens such as zoonotic ones	- Introduced by the use of non-certified and non-hygienic vehicles for transportation	- Application of GHP—bleeding processes, sterilization of bleeding knifes - Use of two-way knife system, where each knife is used for each carcass, then washed and sterilized - Proper slaughter and dressing practices - Opening cut executed from inside to outside during dressing - Dressed carcasses not touching undressed carcasses	[5,10]
		- Chemical - Cleaning chemical residue	- Residues from surfaces after cleaning and sanitation	- Effective cleaning and sanitation programs - Carcasses not touching platforms, and other facilities	[12,13]
		- Physical	- None	- None	
- Further dressing point or abattoir	- Animal dressing at an abattoir or a certified slaughter point - These include the removal of skin, evisceration in preparation for primary meat inspection	- Biological: pathogens, such as zoonotic organisms	- Cross contamination from fecal matter - Poor hygiene application during slaughter - Contamination of dressed carcasses by undressed carcasses	- Training in a good animal slaughter program	[5,10]
		- Chemical:	- Cross contamination from chemicals or oil used in equipment such as air knives (where applicable)	- Equipment maintenance and services - Use of food friendly oil and lubricants - Slaughter training and monitoring	[12,13]
		- Physical	- Bullet particles and bone splinters		[14,17]
- Meat inspection	- Identification of observable abnormalities according to the Meat Safety Act	- Biological: pathogens, such as zoonotic ones	- Animals with zoonotic conditions and pathogens - Poor meat inspection practices	- Trimming and implementation of a good meat inspection program. Secondary inspection for presence of contamination	[5,10]
		- Chemical from equipment at abattoirs	- Poorly maintained equipment	- Trimming all possible or suspected areas of contamination	[12,13]
		- Physical	- Bullet particles and bone splinters	- Removal of bullets particles, pellets and bone splinters	[14,17]
- Treatment and decontamination	- Carcass decontamination plans aimed at reducing the amount or numbers of microorganisms	- Biological: pathogens, such as zoonotic ones	- Cross contamination from treatment solution	- Use sterile decontamination treatment - Change solution in an event that it is contaminated	[5,10]
		- Chemical: chemicals uncertified to be used as treatment solutions	- Contamination from treatment solution	- Proper mixing of a decontamination solution	[12,13]
		- Physical	- None	- Not applicable	
- Chilling	- Final chilling to drop - down the deep bone temperature of carcasses to below 7 °C	- Biological: pathogens found in cooling systems and refrigeration	- Contamination from poor hygiene applications	- Hygiene monitoring at chilling points	[5,10]
		- Chemical:	- Contamination from monitoring processes of temperature reading	- Proper handling of carcasses to avoid contamination	[12,13]
		- Physical - Tips of temperature monitoring probes in carcasses	- Broken thermometers and pH meter tips	- Inspection or measurement of equipment and maintenance	[14,17]

## Data Availability

Data is contained within the article.

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
