# Peer review of "Basic Hazard Control Plan for Small Wild Ungulates Slaughtered for Meat Production"

_foods, 2023, doi:10.3390/foods12071511_

Round 1

Reviewer 1 Report

Manuscript ID: foods-2228431
Type of manuscript: Article
Title: Basic hazards control plan of small wild angulates slaughtered for
meat production purposes

This review study provides an interesting approach about the game animals that are killed or hunted for meat production. The study provided an outline for the rules and the legislations that should be adopted for ensuring the safety of such meat. The following corrections should be considered:

Title: You can delete the word "purposes" from the title

Abstract

Line 21: Please change "; this should be increased by establishing forums" to ", which should be implemented and propagated by establishing forums".

Introduction:

Line 30: The The South African game industry. Delete the extra "The".

Line 44: Correct "transportated" to "transported".

Line 45 : Correct "eviceratio" to "evisceration".

Line 56: Please write the full abbreviation of TPC (the full name must be cited when mentioned for the first time).

Line 57: "the role Organic Acids" correct to "the role of Organic Acids"

Discussion:

Line 280: The health of the farm environment" It would be appropriate to use the word "hygiene" instead of health since it is not related to live animals or human.

Line 375: Change "This thesis" to "This report" or "This review".

Conclusion:

Line 457: "taken in to consideration". Correct to "taken into consideration". 

Author Response

Response to Reviewer 1 Comments

Comment 1

You can delete the word "purposes" from the title.  

Response 1

Thank you, the word purpose has been deleted from the tittle  

Comment 2

Line 21: Please change "; this should be increased by establishing forums" to ", which should be implemented and propagated by establishing forums

Response to comment 2

The change has been implemented as suggested in the abstract and it reads better. 

Comment 3

Line 30: The The South African game industry. Delete the extra "The".

Line 44: Correct "transportated" to "transported".

Line 45 : Correct "eviceratio" to "evisceration".

Line 56: Please write the full abbreviation of TPC (the full name must be cited when mentioned for the first time).

Line 57: "the role Organic Acids" correct to "the role of Organic Acids"

Response to comment 3

“The” has been deleted and the spelling of transported and evisceration and other spelling errors has been fixed across the manuscript.

TPC and all other acronyms are written in full when mentioned the 1st time.

This included the role of organic acids.

Comment 4

Line 280: The health of the farm environment" It would be appropriate to use the word "hygiene" instead of health since it is not related to live animals or human.

Response to comment 4

Thank you for the comment, it is confirmed that reference to farm health has been changed across the manuscript. This was also picked up by other reviewers and the authors have incorporated the suggested changes across the manuscript.

Comments 5

Line 375: Change "This thesis" to "This report" or "This review".

Response to comment 5

Thesis has been replaced with this review and all references to the thesis has been changed accordingly. 

Comments 6

Line 457: "taken in to consideration". Correct to "taken into consideration

Comments and Suggestions for Authors

Response to comment 6

Much appreciated, the spelling of consideration has been changed. The entire document has been proofed by a native English speaker

Reviewer 2 Report

comments are according to row's number:

45: it should be indicated if biosecurity measures are taken, in case of evisceration on the spot, in order to prevent potential diseases spread.

47 to 49: this is apparently in contrast with the concept of "industry"; any "industry" should be aware of hazards and implement corrective/containment measures.

102 to 109: I strongly believe the link of these concept to "animal welfare" is forced and inappropriate.

261; Table 1; I would eliminate the sentence "for the purpose of producing meat; the purpose is "the game",  killing the animals for sport; the animal itself is, unfortunately, a byproduct of "the game".

282: this sentence is an ideological fake; we are not dealing with a particular kind of livestock breeding in order to obtain meat to satisfy basic nutritional needs of a population; we are dealing with "breeding and killing of animals for a game/sport"; and the English name for this activity is self-explanatory.

294 to 306: again, linking these activities with the concept of "animal welfare" simply sound as absurdity.

334: I would skip "and for meat production"; these animals are raised/"produced" for game/sport; not for basic food supply. 

362-362: and infacts, these sentences are fully confirming my doubts above indicated.

402: ideological fake: a  norma/average "consumer" simply accnot afford the expense of 25-30 US$ for 1 minute helicopter flight....

410: acceptability of game is the real purpose of this article; much more than hygiene and safety of this kind of meat production 

GENERAL COMMENT:

I strongly suggest to completely amend this article proposal; focussing ONLY and EXCLUSIVELY about hygiene  hazards and safety of meat for human consumption (from breeding, if wish to do so; to carcass cleaning). Any reference to animal welfare should be avoided. Animal welfare concept and principle cannot stand together with a sport/game finalized to (useless) killing of animals.

Author Response

Response to Reviewer 2 Comments

Comment 1

45: it should be indicated if biosecurity measures are taken, in case of evisceration on the spot, in order to prevent potential diseases spread.

Response 1

Thank you, measures taken to prevent infection during killing has been explained in the manuscript. The explanation included the use of sterilisation of knifes, opening cuts and hygiene compliances.

Comment 2

47 to 49: this is apparently in contrast with the concept of "industry"; any "industry" should be aware of hazards and implement corrective/containment measures

Response to comment 2

Indeed it is in contrast with what is legally expected, more information to highlight these disparities has been included in the manuscript

Comment 3

102 to 109: I strongly believe the link of these concept to "animal welfare" is forced and inappropriate.

Response to comment 3

Thank you, the reference or references to animal welfare is removed in the manuscript.

Comment 4

261; Table 1; I would eliminate the sentence "for the purpose of producing meat; the purpose is "the game",  killing the animals for sport; the animal itself is, unfortunately, a byproduct of "the game".

Response to comment 4

Thank you, while this is indeed correct. In most instances the culling is planned to reduce the numbers of live game in a particular area. Harvested animals are surplus animals that are not deemed suitable for breeding purposes or for hunting as trophies and they must be removed are then typically used for meat production. This point has been clarified in the manuscript. It must be noted that, as a business game meat producers will target the culling season to source game carcasses. 

Comments 5

282: this sentence is an ideological fake; we are not dealing with a particular kind of livestock breeding in order to obtain meat to satisfy basic nutritional needs of a population; we are dealing with "breeding and killing of animals for a game/sport"; and the English name for this activity is self-explanatory.

Response to comment 5

Thank you for the comment. The authors had viewed the control and conservation drivers taken by species’ owners and their strategies to ensure sufficient breeding of game species for meat production. In all these breeding programs, wild game with well-formed horns and sizes are indeed dedicated to trophy hunting. The other subpar or deemed surplus game animals are then culled or harvested for meat production. These processes are generally taking place in the wild, however the farms are vast and are predator free to ensure maximum growth of wild ungulates. I would also like to refer the reviewer to the following manuscripts that shows that breeding of specific species for hunting purposes, does contribute to biodiversity https://doi.org/10.1111/conl.12840 whilst this manuscript deals with consumers’ perceptions of hunted meat https://doi.org/10.1016/j.meatsci.2022.108955

Comment 6

294 to 306: again, linking these activities with the concept of "animal welfare" simply sound as absurdity.

Response to comment 6

Thank you for this comment. All references to animal welfare have been removed across the manuscript.

Comment 7

334: I would skip "and for meat production"; these animals are raised/"produced" for game/sport; not for basic food supply. 

Response to comment 7

Indeed this is correct, however given the high growth of game populations, surplus animals will be culled for meat production. The owners of game farms will also have abattoirs (registered facilities to slaughter and export game meat carcasses). In these cases culled animals will be sent through abattoirs (built for the slaughter of game within the farm). A significant amount of game meat available for export to other game eating countries is from monitored processes of culling not necessarily animals from sports hunting. As an aside, these game ranches/farms also contribute to the upliftment of surrounding communities due to the income that ‘sport hunting’ brings to these communities. I would like to refer the reviewer to the likes of the CAMPFIRE project as an example.

We do respect the Reviewer’s comments (and viewpoint) towards hunting and the breeding of wild species for sport hunting, however, this practise is a reality in many parts of the world and in Africa, it has contributed to not only the wellness of the communities who have to live with the conflict of wildlife-human interactions, but also to biodiversity. That said, we do acknowledge that there are some producers who seem to follow the mantra of “If it pays it stays’ or “If it pays, we will breed it”, unfortunately we live in an economical driven world. We, the authors, do not follow the ideology of these latter producers, however, we are aware that there are surplus animals that enter the food chain and as scientists we are attempting to address a system that will ensure a safe meat product enters into the food chain.

Comment 8

362-362: and infacts, these sentences are fully confirming my doubts above indicated

Response to comment 8

Thank you, we have incorporated in the manuscript information to explain the issue of game animals’ conservation for meat processes. The idea we putting forward is linked to the fact that game conservation is linked with game meat production and it is a big driver to game meat. This manuscript is not about the ethics of these farming practices, but as we have explained above, we are merely looking at the meat value chain from a safety viewpoint.

Comment 9

402: ideological fake: a  norma/average "consumer" simply accnot afford the expense of 25-30 US$ for 1 minute helicopter flight

Response to comment 9

Indeed a normal hunter would not hire a helicopter and it is unethical for a ‘hunter’ to hunt a trophy animal from a helicopter or a vehicle – these practices do not address the idea of “fair chase”. However, with big game conservation facilities the use of a helicopter to swiftly cull sufficient number of animals in a short space of time is a standard practice. Typically, the costs of culling from a helicopter will be covered when around 100 impala (or similar medium sized game animals) are to be harvested. Or around 30-40 larger sized animals such as zebra. Meat from such culling is also subjected to normal hygiene and safety checks by inspectors and authorities. This is different when dealing with hunting and the provision of this meat for private usage by hunters.

In this specific project, the 2-man helicopter belonged to the owner of the property and was regularly used to conduct inspections on the health status of the animals. When needed, the helicopter would be used to cull the required number of animals (around 20) for the days processing in the on-farm abattoir. It typically takes around 30 mins to cull these animals from the helicopter whilst it could take up to 4 hours to cull the same number of animals from a vehicle.

Please note that we use the term cull and not harvest or hunt:

Hunt is the ethical killing of an animal for a specific reason – often sport.

Harvest is the killing of large numbers of animals, where all animals are killed.

Cull is the killing of selected animals within a population.

Both harvest and cull should be done in an ethical manner minimising the stress on the animal being killed and on the remaining animals in the herd.

Comment 10

410: acceptability of game is the real purpose of this article; much more than hygiene and safety of this kind of meat production 

Response to comment 10

Yes that is true and well captured. It is the view of the authors that game meat from all these processes will however enter the market. For these reasons hunters, game facilities and authorities may look at practical measures of ensuring that uncontaminated meat or other processes of ensuring the acceptability of game meat are upheld. The manuscript has been repacked to highlight the importance that cleaner and more hygiene complying processes are upheld by private hunters, conservation facilities and other stakeholders in the game industry. Hence, the developments of a consolidated plan possible to be implemented by ordinary game meat facilities and abattoirs during slaughter of small game meat animals such as Impala or game of the same sizes and killed using the same procedures.

Comment 11

GENERAL COMMENT:

I strongly suggest to completely amend this article proposal; focussing ONLY and EXCLUSIVELY about hygiene  hazards and safety of meat for human consumption (from breeding, if wish to do so; to carcass cleaning). Any reference to animal welfare should be avoided. Animal welfare concept and principle cannot stand together with a sport/game finalized to (useless) killing of animals.

Response to comment 11

Thank you for the guideline. The entire manuscript has been re-written to focus only on hygiene plans from killing to chilling at an abattoir or any game meat establishment. This also incorporates possible measures that could also include carcass cleaning (decontamination). Papers published in the past by the authors also highlighted the responsible usage of decontamination plans. It is the view of the research team that while more emphasis could be spent of cleaner slaughter as a hurdle of note, but carcasses can also be decontaminated (organic acids) to remove or reduce possible organism on the surfaces of the carcasses. We do trust this point has been better packaged and clarifies our view.

Reviewer 3 Report

1. The title does not represent the whole study. This review only focuses on impalas. What does angulates means?

2. May include the market value of wildlife ranching industry in Africa and how this industry can help in food security? Is there any lacking in terms of domestic animal production to support food sustainability for humans?

3. No summary on findings based on previous studies mentioned in Line 54-62

4. No objectives stated in the manuscript

5. Define role-players (line 65). What is the difference between stakeholders?

6. What is FSMS (Line 87)?

7. Define small game killings

8. Methodologies are not clear and comprehensive. What kind of review do you opt?

9. Based on theoretical model & basic meat safety plan, what are your conclusions?

10. No references for the theoretical model and basic meat safety plan under results

11. If you are drafting the hazard identification planning & intervention strategies for small game killing, what are you proposing?

12. The authors do not state the importance of hazard identification in small game killing in regards to meat safety, etc. Provide justifications and research findings to support your problem statement

Author Response

Response to Reviewer 3 Comments

Comment 1

  1. The title does not represent the whole study. This review only focuses on impalas. What does angulates means?

Response 1

Thank you for the comment. The authors have packaged a plan that is generic when handling the three hazards of interest. The focus of the plan is hygiene slaughter so as to ensure that the three major hazards found in meat are reduced or quantified. The tittle clarifies the basic need for different hazards found or introduced by the most used processes of game meat animals culling especially Impala sized. Ungulates are members of the diverse clade Ungulata which primarily consists of large mammals with hooves.

Comment 2

  1. May include the market value of wildlife ranching industry in Africa and how this industry can help in food security? Is there any lacking in terms of domestic animal production to support food sustainability for humans?

Response to comment 2

The growth of game and wild life ranching is documented by many scholars; we have thus explained the steps taken when culling in these ranches. In most game meat producing countries the consumption of game has subsequently increased and this point is clarified in the 1st part of the manuscript. Note that South Africa is in fact a net importer of red meat. It is very difficult to quantify the value of the game meat industry, particularly as some individual animals fetch over US$1 million (https://businesstech.co.za/news/business/112621/this-buffalo-is-worth-more-than-what-top-south-african-ceos-earn-in-5-years/) – obviously these animals illnot be hunted for meat. However, their offspring that do not meet the required standards (be it hrons, sex, or pelt colour) are often harvested for meat – at a lower value. Also there is no data base in South Africa indicating the number of wild ungulates in the country – in fact data is only known for the formal commercial agriculture  sectors as pertaining to the guesstimate of the livestock numbers, this does not include numbers for livestock farmed in the informal sector.

Comment 3

  1. No summary on findings based on previous studies mentioned in Line 54-62ate.

Response to comment 3

Thank you, findings of the previous studies are summarised and captured in the manuscript especially the last part of the introduction. The findings have made it necessary to develop this plan for meat production. To also be precise and clearer, we have summed up the finding and made references to the different conclusions of published work on these issues. 

Comment 4

  1. No objectives stated in the manuscript

Response to comment 4

The rewritten manuscript has included the objective in the last paragraph of the second page and incorporated this information to explain the need for such a plan especially in game meat processes of slaughter and meat production.

Comments 5

  1. Define role-players (line 65). What is the difference between stakeholders?

Response to comment 5

The stakeholders and role players could be described as people or organisations that have a meaningful role in processes linked with game meat production. These organisations and their role have been captured in the discussions and reference to the suggested model for game meat production in the manuscript. 

Comment 6

  1. What is FSMS (Line 87)?

Thank you, FSMS is Food Safety Management System. This has been clarified and all acronyms have been written in full the 1st time they appear in the manuscript.   

294 to 306: again, linking these activities with the concept of "animal welfare" simply sound as absurdity.

Response to comment 6

Thank you, this is similar to the comments made by other reviewers and reference to welfare of the animal has been removed across the manuscript.

Comment 7

Define small game killings

Response to comment 7

The description of small game and examples thereof has been included in the manuscript – ‘small game’ is also defined in the Meat Safety Act (https://www.gov.za/sites/default/files/gcis_document/201409/a40-000.pdf)

Veterinary Procedure (VPN) for registration of hunters and game farmer/s

(http://www.old.dalrrd.gov.za/vetweb/VPN%20&%20SOP/008-VPN%2008%20Standard%20for%20the%20registration%20of%20hunters%20for%20harvesting%20wild%20game%20intended%20for%20export%20of%20game%20meat..pdf)

Comment 8

  1. Methodologies are not clear and comprehensive. What kind of review do you opt?

Response to comment 8

This paper follows a narrative review approach. This review stems on from 6 PhD papers that have been published: 2 papers looked at each possible hazard (Microbiological – Total Plate Count, Coliform, Ecoli and Salmonella); Chemical hazards (Lead, Cadmium, Arsenic and Mercury) and Physical hazards (Ammunition particles dispersion and bones splinters). This 7th paper combines all the efforts and thus identifies the key stakeholders in the industry and develops a basic plan for hygiene control of small game slaughter. A reviewer form one of the earlier papers suggested that such an encompassing paper describing this plan would be welcome.

Comment 9

  1. Based on theoretical model & basic meat safety plan, what are your conclusions?

Response to comment 9

The general fragmentation of hygiene controls, activities and actions by different role players has led to the proposal that game meat animal slaughter stakeholders must come together in a forum and discuss issues linked with these processes. The authors used published articles to develop a basic hygiene plan that could be discussed in this forum and implemented as needed. This information is included in the manuscript.

Comment 10

  1. No references for the theoretical model and basic meat safety plan under results

Response to comment 10

The concept of the model is developed and packaged by the authors, this is similar to the basic plan.

Comment 11

  1. If you are drafting the hazard identification planning & intervention strategies for small game killing, what are you proposing?

Response to comment 11

Thank you, the proposal being carried out is that game slaughter or stakeholders as identified, must follow the guidelines and identify the needed interventions to enable the production of safer meat. In this forum, training, refreshers and updates on regulations will be done, this forum could be used as information sharing platform for personal development. The basic plan will then be implemented during the actual killing or slaughter of small game (especially the control column)

Comment 12

  1. The authors do not state the importance of hazard identification in small game killing in regards to meat safety, etc. Provide justifications and research findings to support your problem state

Response to comment 12

Thank you, in the manuscript the risks of consuming hazards are clarified. The authors have also included not only hazards identification but also possible controls implementable during the slaughter of these animals

Response to Reviewer 3 Comments

Comment 1

  1. The title does not represent the whole study. This review only focuses on impalas. What does angulates means?

Response 1

Thank you for the comment. The authors have packaged a plan that is generic when handling the three hazards of interest. The focus of the plan is hygiene slaughter so as to ensure that the three major hazards found in meat are reduced or quantified. The tittle clarifies the basic need for different hazards found or introduced by the most used processes of game meat animals culling especially Impala sized. Ungulates are members of the diverse clade Ungulata which primarily consists of large mammals with hooves.

Comment 2

  1. May include the market value of wildlife ranching industry in Africa and how this industry can help in food security? Is there any lacking in terms of domestic animal production to support food sustainability for humans?

Response to comment 2

The growth of game and wild life ranching is documented by many scholars; we have thus explained the steps taken when culling in these ranches. In most game meat producing countries the consumption of game has subsequently increased and this point is clarified in the 1st part of the manuscript. Note that South Africa is in fact a net importer of red meat. It is very difficult to quantify the value of the game meat industry, particularly as some individual animals fetch over US$1 million (https://businesstech.co.za/news/business/112621/this-buffalo-is-worth-more-than-what-top-south-african-ceos-earn-in-5-years/) – obviously these animals illnot be hunted for meat. However, their offspring that do not meet the required standards (be it hrons, sex, or pelt colour) are often harvested for meat – at a lower value. Also there is no data base in South Africa indicating the number of wild ungulates in the country – in fact data is only known for the formal commercial agriculture  sectors as pertaining to the guesstimate of the livestock numbers, this does not include numbers for livestock farmed in the informal sector.

Comment 3

  1. No summary on findings based on previous studies mentioned in Line 54-62ate.

Response to comment 3

Thank you, findings of the previous studies are summarised and captured in the manuscript especially the last part of the introduction. The findings have made it necessary to develop this plan for meat production. To also be precise and clearer, we have summed up the finding and made references to the different conclusions of published work on these issues. 

Comment 4

  1. No objectives stated in the manuscript

Response to comment 4

The rewritten manuscript has included the objective in the last paragraph of the second page and incorporated this information to explain the need for such a plan especially in game meat processes of slaughter and meat production.

Comments 5

  1. Define role-players (line 65). What is the difference between stakeholders?

Response to comment 5

The stakeholders and role players could be described as people or organisations that have a meaningful role in processes linked with game meat production. These organisations and their role have been captured in the discussions and reference to the suggested model for game meat production in the manuscript. 

Comment 6

  1. What is FSMS (Line 87)?

Thank you, FSMS is Food Safety Management System. This has been clarified and all acronyms have been written in full the 1st time they appear in the manuscript.   

294 to 306: again, linking these activities with the concept of "animal welfare" simply sound as absurdity.

Response to comment 6

Thank you, this is similar to the comments made by other reviewers and reference to welfare of the animal has been removed across the manuscript.

Comment 7

Define small game killings

Response to comment 7

The description of small game and examples thereof has been included in the manuscript – ‘small game’ is also defined in the Meat Safety Act (https://www.gov.za/sites/default/files/gcis_document/201409/a40-000.pdf)

Veterinary Procedure (VPN) for registration of hunters and game farmer/s

(http://www.old.dalrrd.gov.za/vetweb/VPN%20&%20SOP/008-VPN%2008%20Standard%20for%20the%20registration%20of%20hunters%20for%20harvesting%20wild%20game%20intended%20for%20export%20of%20game%20meat..pdf)

Comment 8

  1. Methodologies are not clear and comprehensive. What kind of review do you opt?

Response to comment 8

This paper follows a narrative review approach. This review stems on from 6 PhD papers that have been published: 2 papers looked at each possible hazard (Microbiological – Total Plate Count, Coliform, Ecoli and Salmonella); Chemical hazards (Lead, Cadmium, Arsenic and Mercury) and Physical hazards (Ammunition particles dispersion and bones splinters). This 7th paper combines all the efforts and thus identifies the key stakeholders in the industry and develops a basic plan for hygiene control of small game slaughter. A reviewer form one of the earlier papers suggested that such an encompassing paper describing this plan would be welcome.

Comment 9

  1. Based on theoretical model & basic meat safety plan, what are your conclusions?

Response to comment 9

The general fragmentation of hygiene controls, activities and actions by different role players has led to the proposal that game meat animal slaughter stakeholders must come together in a forum and discuss issues linked with these processes. The authors used published articles to develop a basic hygiene plan that could be discussed in this forum and implemented as needed. This information is included in the manuscript.

Comment 10

  1. No references for the theoretical model and basic meat safety plan under results

Response to comment 10

The concept of the model is developed and packaged by the authors, this is similar to the basic plan.

Comment 11

  1. If you are drafting the hazard identification planning & intervention strategies for small game killing, what are you proposing?

Response to comment 11

Thank you, the proposal being carried out is that game slaughter or stakeholders as identified, must follow the guidelines and identify the needed interventions to enable the production of safer meat. In this forum, training, refreshers and updates on regulations will be done, this forum could be used as information sharing platform for personal development. The basic plan will then be implemented during the actual killing or slaughter of small game (especially the control column)

Comment 12

  1. The authors do not state the importance of hazard identification in small game killing in regards to meat safety, etc. Provide justifications and research findings to support your problem state

Response to comment 12

Thank you, in the manuscript the risks of consuming hazards are clarified. The authors have also included not only hazards identification but also possible controls implementable during the slaughter of these animals

Response to Reviewer 3 Comments

Comment 1

  1. The title does not represent the whole study. This review only focuses on impalas. What does angulates means?

Response 1

Thank you for the comment. The authors have packaged a plan that is generic when handling the three hazards of interest. The focus of the plan is hygiene slaughter so as to ensure that the three major hazards found in meat are reduced or quantified. The tittle clarifies the basic need for different hazards found or introduced by the most used processes of game meat animals culling especially Impala sized. Ungulates are members of the diverse clade Ungulata which primarily consists of large mammals with hooves.

Comment 2

  1. May include the market value of wildlife ranching industry in Africa and how this industry can help in food security? Is there any lacking in terms of domestic animal production to support food sustainability for humans?

Response to comment 2

The growth of game and wild life ranching is documented by many scholars; we have thus explained the steps taken when culling in these ranches. In most game meat producing countries the consumption of game has subsequently increased and this point is clarified in the 1st part of the manuscript. Note that South Africa is in fact a net importer of red meat. It is very difficult to quantify the value of the game meat industry, particularly as some individual animals fetch over US$1 million (https://businesstech.co.za/news/business/112621/this-buffalo-is-worth-more-than-what-top-south-african-ceos-earn-in-5-years/) – obviously these animals illnot be hunted for meat. However, their offspring that do not meet the required standards (be it hrons, sex, or pelt colour) are often harvested for meat – at a lower value. Also there is no data base in South Africa indicating the number of wild ungulates in the country – in fact data is only known for the formal commercial agriculture  sectors as pertaining to the guesstimate of the livestock numbers, this does not include numbers for livestock farmed in the informal sector.

Comment 3

  1. No summary on findings based on previous studies mentioned in Line 54-62ate.

Response to comment 3

Thank you, findings of the previous studies are summarised and captured in the manuscript especially the last part of the introduction. The findings have made it necessary to develop this plan for meat production. To also be precise and clearer, we have summed up the finding and made references to the different conclusions of published work on these issues. 

Comment 4

  1. No objectives stated in the manuscript

Response to comment 4

The rewritten manuscript has included the objective in the last paragraph of the second page and incorporated this information to explain the need for such a plan especially in game meat processes of slaughter and meat production.

Comments 5

  1. Define role-players (line 65). What is the difference between stakeholders?

Response to comment 5

The stakeholders and role players could be described as people or organisations that have a meaningful role in processes linked with game meat production. These organisations and their role have been captured in the discussions and reference to the suggested model for game meat production in the manuscript. 

Comment 6

  1. What is FSMS (Line 87)?

Thank you, FSMS is Food Safety Management System. This has been clarified and all acronyms have been written in full the 1st time they appear in the manuscript.   

294 to 306: again, linking these activities with the concept of "animal welfare" simply sound as absurdity.

Response to comment 6

Thank you, this is similar to the comments made by other reviewers and reference to welfare of the animal has been removed across the manuscript.

Comment 7

Define small game killings

Response to comment 7

The description of small game and examples thereof has been included in the manuscript – ‘small game’ is also defined in the Meat Safety Act (https://www.gov.za/sites/default/files/gcis_document/201409/a40-000.pdf)

Veterinary Procedure (VPN) for registration of hunters and game farmer/s

(http://www.old.dalrrd.gov.za/vetweb/VPN%20&%20SOP/008-VPN%2008%20Standard%20for%20the%20registration%20of%20hunters%20for%20harvesting%20wild%20game%20intended%20for%20export%20of%20game%20meat..pdf)

Comment 8

  1. Methodologies are not clear and comprehensive. What kind of review do you opt?

Response to comment 8

This paper follows a narrative review approach. This review stems on from 6 PhD papers that have been published: 2 papers looked at each possible hazard (Microbiological – Total Plate Count, Coliform, Ecoli and Salmonella); Chemical hazards (Lead, Cadmium, Arsenic and Mercury) and Physical hazards (Ammunition particles dispersion and bones splinters). This 7th paper combines all the efforts and thus identifies the key stakeholders in the industry and develops a basic plan for hygiene control of small game slaughter. A reviewer form one of the earlier papers suggested that such an encompassing paper describing this plan would be welcome.

Comment 9

  1. Based on theoretical model & basic meat safety plan, what are your conclusions?

Response to comment 9

The general fragmentation of hygiene controls, activities and actions by different role players has led to the proposal that game meat animal slaughter stakeholders must come together in a forum and discuss issues linked with these processes. The authors used published articles to develop a basic hygiene plan that could be discussed in this forum and implemented as needed. This information is included in the manuscript.

Comment 10

  1. No references for the theoretical model and basic meat safety plan under results

Response to comment 10

The concept of the model is developed and packaged by the authors, this is similar to the basic plan.

Comment 11

  1. If you are drafting the hazard identification planning & intervention strategies for small game killing, what are you proposing?

Response to comment 11

Thank you, the proposal being carried out is that game slaughter or stakeholders as identified, must follow the guidelines and identify the needed interventions to enable the production of safer meat. In this forum, training, refreshers and updates on regulations will be done, this forum could be used as information sharing platform for personal development. The basic plan will then be implemented during the actual killing or slaughter of small game (especially the control column)

Comment 12

  1. The authors do not state the importance of hazard identification in small game killing in regards to meat safety, etc. Provide justifications and research findings to support your problem state

Response to comment 12

Thank you, in the manuscript the risks of consuming hazards are clarified. The authors have also included not only hazards identification but also possible controls implementable during the slaughter of these animals

Response to Reviewer 3 Comments

Comment 1

  1. The title does not represent the whole study. This review only focuses on impalas. What does angulates means?

Response 1

Thank you for the comment. The authors have packaged a plan that is generic when handling the three hazards of interest. The focus of the plan is hygiene slaughter so as to ensure that the three major hazards found in meat are reduced or quantified. The tittle clarifies the basic need for different hazards found or introduced by the most used processes of game meat animals culling especially Impala sized. Ungulates are members of the diverse clade Ungulata which primarily consists of large mammals with hooves.

Comment 2

  1. May include the market value of wildlife ranching industry in Africa and how this industry can help in food security? Is there any lacking in terms of domestic animal production to support food sustainability for humans?

Response to comment 2

The growth of game and wild life ranching is documented by many scholars; we have thus explained the steps taken when culling in these ranches. In most game meat producing countries the consumption of game has subsequently increased and this point is clarified in the 1st part of the manuscript. Note that South Africa is in fact a net importer of red meat. It is very difficult to quantify the value of the game meat industry, particularly as some individual animals fetch over US$1 million (https://businesstech.co.za/news/business/112621/this-buffalo-is-worth-more-than-what-top-south-african-ceos-earn-in-5-years/) – obviously these animals illnot be hunted for meat. However, their offspring that do not meet the required standards (be it hrons, sex, or pelt colour) are often harvested for meat – at a lower value. Also there is no data base in South Africa indicating the number of wild ungulates in the country – in fact data is only known for the formal commercial agriculture  sectors as pertaining to the guesstimate of the livestock numbers, this does not include numbers for livestock farmed in the informal sector.

Comment 3

  1. No summary on findings based on previous studies mentioned in Line 54-62ate.

Response to comment 3

Thank you, findings of the previous studies are summarised and captured in the manuscript especially the last part of the introduction. The findings have made it necessary to develop this plan for meat production. To also be precise and clearer, we have summed up the finding and made references to the different conclusions of published work on these issues. 

Comment 4

  1. No objectives stated in the manuscript

Response to comment 4

The rewritten manuscript has included the objective in the last paragraph of the second page and incorporated this information to explain the need for such a plan especially in game meat processes of slaughter and meat production.

Comments 5

  1. Define role-players (line 65). What is the difference between stakeholders?

Response to comment 5

The stakeholders and role players could be described as people or organisations that have a meaningful role in processes linked with game meat production. These organisations and their role have been captured in the discussions and reference to the suggested model for game meat production in the manuscript. 

Comment 6

  1. What is FSMS (Line 87)?

Thank you, FSMS is Food Safety Management System. This has been clarified and all acronyms have been written in full the 1st time they appear in the manuscript.   

294 to 306: again, linking these activities with the concept of "animal welfare" simply sound as absurdity.

Response to comment 6

Thank you, this is similar to the comments made by other reviewers and reference to welfare of the animal has been removed across the manuscript.

Comment 7

Define small game killings

Response to comment 7

The description of small game and examples thereof has been included in the manuscript – ‘small game’ is also defined in the Meat Safety Act (https://www.gov.za/sites/default/files/gcis_document/201409/a40-000.pdf)

Veterinary Procedure (VPN) for registration of hunters and game farmer/s

(http://www.old.dalrrd.gov.za/vetweb/VPN%20&%20SOP/008-VPN%2008%20Standard%20for%20the%20registration%20of%20hunters%20for%20harvesting%20wild%20game%20intended%20for%20export%20of%20game%20meat..pdf)

Comment 8

  1. Methodologies are not clear and comprehensive. What kind of review do you opt?

Response to comment 8

This paper follows a narrative review approach. This review stems on from 6 PhD papers that have been published: 2 papers looked at each possible hazard (Microbiological – Total Plate Count, Coliform, Ecoli and Salmonella); Chemical hazards (Lead, Cadmium, Arsenic and Mercury) and Physical hazards (Ammunition particles dispersion and bones splinters). This 7th paper combines all the efforts and thus identifies the key stakeholders in the industry and develops a basic plan for hygiene control of small game slaughter. A reviewer form one of the earlier papers suggested that such an encompassing paper describing this plan would be welcome.

Comment 9

  1. Based on theoretical model & basic meat safety plan, what are your conclusions?

Response to comment 9

The general fragmentation of hygiene controls, activities and actions by different role players has led to the proposal that game meat animal slaughter stakeholders must come together in a forum and discuss issues linked with these processes. The authors used published articles to develop a basic hygiene plan that could be discussed in this forum and implemented as needed. This information is included in the manuscript.

Comment 10

  1. No references for the theoretical model and basic meat safety plan under results

Response to comment 10

The concept of the model is developed and packaged by the authors, this is similar to the basic plan.

Comment 11

  1. If you are drafting the hazard identification planning & intervention strategies for small game killing, what are you proposing?

Response to comment 11

Thank you, the proposal being carried out is that game slaughter or stakeholders as identified, must follow the guidelines and identify the needed interventions to enable the production of safer meat. In this forum, training, refreshers and updates on regulations will be done, this forum could be used as information sharing platform for personal development. The basic plan will then be implemented during the actual killing or slaughter of small game (especially the control column)

Comment 12

  1. The authors do not state the importance of hazard identification in small game killing in regards to meat safety, etc. Provide justifications and research findings to support your problem state

Response to comment 12

Thank you, in the manuscript the risks of consuming hazards are clarified. The authors have also included not only hazards identification but also possible controls implementable during the slaughter of these animals

Response to Reviewer 3 Comments

Comment 1

  1. The title does not represent the whole study. This review only focuses on impalas. What does angulates means?

Response 1

Thank you for the comment. The authors have packaged a plan that is generic when handling the three hazards of interest. The focus of the plan is hygiene slaughter so as to ensure that the three major hazards found in meat are reduced or quantified. The tittle clarifies the basic need for different hazards found or introduced by the most used processes of game meat animals culling especially Impala sized. Ungulates are members of the diverse clade Ungulata which primarily consists of large mammals with hooves.

Comment 2

  1. May include the market value of wildlife ranching industry in Africa and how this industry can help in food security? Is there any lacking in terms of domestic animal production to support food sustainability for humans?

Response to comment 2

The growth of game and wild life ranching is documented by many scholars; we have thus explained the steps taken when culling in these ranches. In most game meat producing countries the consumption of game has subsequently increased and this point is clarified in the 1st part of the manuscript. Note that South Africa is in fact a net importer of red meat. It is very difficult to quantify the value of the game meat industry, particularly as some individual animals fetch over US$1 million (https://businesstech.co.za/news/business/112621/this-buffalo-is-worth-more-than-what-top-south-african-ceos-earn-in-5-years/) – obviously these animals illnot be hunted for meat. However, their offspring that do not meet the required standards (be it hrons, sex, or pelt colour) are often harvested for meat – at a lower value. Also there is no data base in South Africa indicating the number of wild ungulates in the country – in fact data is only known for the formal commercial agriculture  sectors as pertaining to the guesstimate of the livestock numbers, this does not include numbers for livestock farmed in the informal sector.

Comment 3

  1. No summary on findings based on previous studies mentioned in Line 54-62ate.

Response to comment 3

Thank you, findings of the previous studies are summarised and captured in the manuscript especially the last part of the introduction. The findings have made it necessary to develop this plan for meat production. To also be precise and clearer, we have summed up the finding and made references to the different conclusions of published work on these issues. 

Comment 4

  1. No objectives stated in the manuscript

Response to comment 4

The rewritten manuscript has included the objective in the last paragraph of the second page and incorporated this information to explain the need for such a plan especially in game meat processes of slaughter and meat production.

Comments 5

  1. Define role-players (line 65). What is the difference between stakeholders?

Response to comment 5

The stakeholders and role players could be described as people or organisations that have a meaningful role in processes linked with game meat production. These organisations and their role have been captured in the discussions and reference to the suggested model for game meat production in the manuscript. 

Comment 6

  1. What is FSMS (Line 87)?

Thank you, FSMS is Food Safety Management System. This has been clarified and all acronyms have been written in full the 1st time they appear in the manuscript.   

294 to 306: again, linking these activities with the concept of "animal welfare" simply sound as absurdity.

Response to comment 6

Thank you, this is similar to the comments made by other reviewers and reference to welfare of the animal has been removed across the manuscript.

Comment 7

Define small game killings

Response to comment 7

The description of small game and examples thereof has been included in the manuscript – ‘small game’ is also defined in the Meat Safety Act (https://www.gov.za/sites/default/files/gcis_document/201409/a40-000.pdf)

Veterinary Procedure (VPN) for registration of hunters and game farmer/s

(http://www.old.dalrrd.gov.za/vetweb/VPN%20&%20SOP/008-VPN%2008%20Standard%20for%20the%20registration%20of%20hunters%20for%20harvesting%20wild%20game%20intended%20for%20export%20of%20game%20meat..pdf)

Comment 8

  1. Methodologies are not clear and comprehensive. What kind of review do you opt?

Response to comment 8

This paper follows a narrative review approach. This review stems on from 6 PhD papers that have been published: 2 papers looked at each possible hazard (Microbiological – Total Plate Count, Coliform, Ecoli and Salmonella); Chemical hazards (Lead, Cadmium, Arsenic and Mercury) and Physical hazards (Ammunition particles dispersion and bones splinters). This 7th paper combines all the efforts and thus identifies the key stakeholders in the industry and develops a basic plan for hygiene control of small game slaughter. A reviewer form one of the earlier papers suggested that such an encompassing paper describing this plan would be welcome.

Comment 9

  1. Based on theoretical model & basic meat safety plan, what are your conclusions?

Response to comment 9

The general fragmentation of hygiene controls, activities and actions by different role players has led to the proposal that game meat animal slaughter stakeholders must come together in a forum and discuss issues linked with these processes. The authors used published articles to develop a basic hygiene plan that could be discussed in this forum and implemented as needed. This information is included in the manuscript.

Comment 10

  1. No references for the theoretical model and basic meat safety plan under results

Response to comment 10

The concept of the model is developed and packaged by the authors, this is similar to the basic plan.

Comment 11

  1. If you are drafting the hazard identification planning & intervention strategies for small game killing, what are you proposing?

Response to comment 11

Thank you, the proposal being carried out is that game slaughter or stakeholders as identified, must follow the guidelines and identify the needed interventions to enable the production of safer meat. In this forum, training, refreshers and updates on regulations will be done, this forum could be used as information sharing platform for personal development. The basic plan will then be implemented during the actual killing or slaughter of small game (especially the control column)

Comment 12

  1. The authors do not state the importance of hazard identification in small game killing in regards to meat safety, etc. Provide justifications and research findings to support your problem state

Response to comment 12

Thank you, in the manuscript the risks of consuming hazards are clarified. The authors have also included not only hazards identification but also possible controls implementable during the slaughter of these animals

Response to Reviewer 3 Comments

Comment 1

  1. The title does not represent the whole study. This review only focuses on impalas. What does angulates means?

Response 1

Thank you for the comment. The authors have packaged a plan that is generic when handling the three hazards of interest. The focus of the plan is hygiene slaughter so as to ensure that the three major hazards found in meat are reduced or quantified. The tittle clarifies the basic need for different hazards found or introduced by the most used processes of game meat animals culling especially Impala sized. Ungulates are members of the diverse clade Ungulata which primarily consists of large mammals with hooves.

Comment 2

  1. May include the market value of wildlife ranching industry in Africa and how this industry can help in food security? Is there any lacking in terms of domestic animal production to support food sustainability for humans?

Response to comment 2

The growth of game and wild life ranching is documented by many scholars; we have thus explained the steps taken when culling in these ranches. In most game meat producing countries the consumption of game has subsequently increased and this point is clarified in the 1st part of the manuscript. Note that South Africa is in fact a net importer of red meat. It is very difficult to quantify the value of the game meat industry, particularly as some individual animals fetch over US$1 million (https://businesstech.co.za/news/business/112621/this-buffalo-is-worth-more-than-what-top-south-african-ceos-earn-in-5-years/) – obviously these animals illnot be hunted for meat. However, their offspring that do not meet the required standards (be it hrons, sex, or pelt colour) are often harvested for meat – at a lower value. Also there is no data base in South Africa indicating the number of wild ungulates in the country – in fact data is only known for the formal commercial agriculture  sectors as pertaining to the guesstimate of the livestock numbers, this does not include numbers for livestock farmed in the informal sector.

Comment 3

  1. No summary on findings based on previous studies mentioned in Line 54-62ate.

Response to comment 3

Thank you, findings of the previous studies are summarised and captured in the manuscript especially the last part of the introduction. The findings have made it necessary to develop this plan for meat production. To also be precise and clearer, we have summed up the finding and made references to the different conclusions of published work on these issues. 

Comment 4

  1. No objectives stated in the manuscript

Response to comment 4

The rewritten manuscript has included the objective in the last paragraph of the second page and incorporated this information to explain the need for such a plan especially in game meat processes of slaughter and meat production.

Comments 5

  1. Define role-players (line 65). What is the difference between stakeholders?

Response to comment 5

The stakeholders and role players could be described as people or organisations that have a meaningful role in processes linked with game meat production. These organisations and their role have been captured in the discussions and reference to the suggested model for game meat production in the manuscript. 

Comment 6

  1. What is FSMS (Line 87)?

Thank you, FSMS is Food Safety Management System. This has been clarified and all acronyms have been written in full the 1st time they appear in the manuscript.   

294 to 306: again, linking these activities with the concept of "animal welfare" simply sound as absurdity.

Response to comment 6

Thank you, this is similar to the comments made by other reviewers and reference to welfare of the animal has been removed across the manuscript.

Comment 7

Define small game killings

Response to comment 7

The description of small game and examples thereof has been included in the manuscript – ‘small game’ is also defined in the Meat Safety Act (https://www.gov.za/sites/default/files/gcis_document/201409/a40-000.pdf)

Veterinary Procedure (VPN) for registration of hunters and game farmer/s

(http://www.old.dalrrd.gov.za/vetweb/VPN%20&%20SOP/008-VPN%2008%20Standard%20for%20the%20registration%20of%20hunters%20for%20harvesting%20wild%20game%20intended%20for%20export%20of%20game%20meat..pdf)

Comment 8

  1. Methodologies are not clear and comprehensive. What kind of review do you opt?

Response to comment 8

This paper follows a narrative review approach. This review stems on from 6 PhD papers that have been published: 2 papers looked at each possible hazard (Microbiological – Total Plate Count, Coliform, Ecoli and Salmonella); Chemical hazards (Lead, Cadmium, Arsenic and Mercury) and Physical hazards (Ammunition particles dispersion and bones splinters). This 7th paper combines all the efforts and thus identifies the key stakeholders in the industry and develops a basic plan for hygiene control of small game slaughter. A reviewer form one of the earlier papers suggested that such an encompassing paper describing this plan would be welcome.

Comment 9

  1. Based on theoretical model & basic meat safety plan, what are your conclusions?

Response to comment 9

The general fragmentation of hygiene controls, activities and actions by different role players has led to the proposal that game meat animal slaughter stakeholders must come together in a forum and discuss issues linked with these processes. The authors used published articles to develop a basic hygiene plan that could be discussed in this forum and implemented as needed. This information is included in the manuscript.

Comment 10

  1. No references for the theoretical model and basic meat safety plan under results

Response to comment 10

The concept of the model is developed and packaged by the authors, this is similar to the basic plan.

Comment 11

  1. If you are drafting the hazard identification planning & intervention strategies for small game killing, what are you proposing?

Response to comment 11

Thank you, the proposal being carried out is that game slaughter or stakeholders as identified, must follow the guidelines and identify the needed interventions to enable the production of safer meat. In this forum, training, refreshers and updates on regulations will be done, this forum could be used as information sharing platform for personal development. The basic plan will then be implemented during the actual killing or slaughter of small game (especially the control column)

Comment 12

  1. The authors do not state the importance of hazard identification in small game killing in regards to meat safety, etc. Provide justifications and research findings to support your problem state

Response to comment 12

Thank you, in the manuscript the risks of consuming hazards are clarified. The authors have also included not only hazards identification but also possible controls implementable during the slaughter of these animals

Response to Reviewer 3 Comments

Comment 1

  1. The title does not represent the whole study. This review only focuses on impalas. What does angulates means?

Response 1

Thank you for the comment. The authors have packaged a plan that is generic when handling the three hazards of interest. The focus of the plan is hygiene slaughter so as to ensure that the three major hazards found in meat are reduced or quantified. The tittle clarifies the basic need for different hazards found or introduced by the most used processes of game meat animals culling especially Impala sized. Ungulates are members of the diverse clade Ungulata which primarily consists of large mammals with hooves.

Comment 2

  1. May include the market value of wildlife ranching industry in Africa and how this industry can help in food security? Is there any lacking in terms of domestic animal production to support food sustainability for humans?

Response to comment 2

The growth of game and wild life ranching is documented by many scholars; we have thus explained the steps taken when culling in these ranches. In most game meat producing countries the consumption of game has subsequently increased and this point is clarified in the 1st part of the manuscript. Note that South Africa is in fact a net importer of red meat. It is very difficult to quantify the value of the game meat industry, particularly as some individual animals fetch over US$1 million (https://businesstech.co.za/news/business/112621/this-buffalo-is-worth-more-than-what-top-south-african-ceos-earn-in-5-years/) – obviously these animals illnot be hunted for meat. However, their offspring that do not meet the required standards (be it hrons, sex, or pelt colour) are often harvested for meat – at a lower value. Also there is no data base in South Africa indicating the number of wild ungulates in the country – in fact data is only known for the formal commercial agriculture  sectors as pertaining to the guesstimate of the livestock numbers, this does not include numbers for livestock farmed in the informal sector.

Comment 3

  1. No summary on findings based on previous studies mentioned in Line 54-62ate.

Response to comment 3

Thank you, findings of the previous studies are summarised and captured in the manuscript especially the last part of the introduction. The findings have made it necessary to develop this plan for meat production. To also be precise and clearer, we have summed up the finding and made references to the different conclusions of published work on these issues. 

Comment 4

  1. No objectives stated in the manuscript

Response to comment 4

The rewritten manuscript has included the objective in the last paragraph of the second page and incorporated this information to explain the need for such a plan especially in game meat processes of slaughter and meat production.

Comments 5

  1. Define role-players (line 65). What is the difference between stakeholders?

Response to comment 5

The stakeholders and role players could be described as people or organisations that have a meaningful role in processes linked with game meat production. These organisations and their role have been captured in the discussions and reference to the suggested model for game meat production in the manuscript. 

Comment 6

  1. What is FSMS (Line 87)?

Thank you, FSMS is Food Safety Management System. This has been clarified and all acronyms have been written in full the 1st time they appear in the manuscript.   

294 to 306: again, linking these activities with the concept of "animal welfare" simply sound as absurdity.

Response to comment 6

Thank you, this is similar to the comments made by other reviewers and reference to welfare of the animal has been removed across the manuscript.

Comment 7

Define small game killings

Response to comment 7

The description of small game and examples thereof has been included in the manuscript – ‘small game’ is also defined in the Meat Safety Act (https://www.gov.za/sites/default/files/gcis_document/201409/a40-000.pdf)

Veterinary Procedure (VPN) for registration of hunters and game farmer/s

(http://www.old.dalrrd.gov.za/vetweb/VPN%20&%20SOP/008-VPN%2008%20Standard%20for%20the%20registration%20of%20hunters%20for%20harvesting%20wild%20game%20intended%20for%20export%20of%20game%20meat..pdf)

Comment 8

  1. Methodologies are not clear and comprehensive. What kind of review do you opt?

Response to comment 8

This paper follows a narrative review approach. This review stems on from 6 PhD papers that have been published: 2 papers looked at each possible hazard (Microbiological – Total Plate Count, Coliform, Ecoli and Salmonella); Chemical hazards (Lead, Cadmium, Arsenic and Mercury) and Physical hazards (Ammunition particles dispersion and bones splinters). This 7th paper combines all the efforts and thus identifies the key stakeholders in the industry and develops a basic plan for hygiene control of small game slaughter. A reviewer form one of the earlier papers suggested that such an encompassing paper describing this plan would be welcome.

Comment 9

  1. Based on theoretical model & basic meat safety plan, what are your conclusions?

Response to comment 9

The general fragmentation of hygiene controls, activities and actions by different role players has led to the proposal that game meat animal slaughter stakeholders must come together in a forum and discuss issues linked with these processes. The authors used published articles to develop a basic hygiene plan that could be discussed in this forum and implemented as needed. This information is included in the manuscript.

Comment 10

  1. No references for the theoretical model and basic meat safety plan under results

Response to comment 10

The concept of the model is developed and packaged by the authors, this is similar to the basic plan.

Comment 11

  1. If you are drafting the hazard identification planning & intervention strategies for small game killing, what are you proposing?

Response to comment 11

Thank you, the proposal being carried out is that game slaughter or stakeholders as identified, must follow the guidelines and identify the needed interventions to enable the production of safer meat. In this forum, training, refreshers and updates on regulations will be done, this forum could be used as information sharing platform for personal development. The basic plan will then be implemented during the actual killing or slaughter of small game (especially the control column)

Comment 12

  1. The authors do not state the importance of hazard identification in small game killing in regards to meat safety, etc. Provide justifications and research findings to support your problem state

Response to comment 12

Thank you, in the manuscript the risks of consuming hazards are clarified. The authors have also included not only hazards identification but also possible controls implementable during the slaughter of these animals

Response to Reviewer 3 Comments

Comment 1

  1. The title does not represent the whole study. This review only focuses on impalas. What does angulates means?

Response 1

Thank you for the comment. The authors have packaged a plan that is generic when handling the three hazards of interest. The focus of the plan is hygiene slaughter so as to ensure that the three major hazards found in meat are reduced or quantified. The tittle clarifies the basic need for different hazards found or introduced by the most used processes of game meat animals culling especially Impala sized. Ungulates are members of the diverse clade Ungulata which primarily consists of large mammals with hooves.

Comment 2

  1. May include the market value of wildlife ranching industry in Africa and how this industry can help in food security? Is there any lacking in terms of domestic animal production to support food sustainability for humans?

Response to comment 2

The growth of game and wild life ranching is documented by many scholars; we have thus explained the steps taken when culling in these ranches. In most game meat producing countries the consumption of game has subsequently increased and this point is clarified in the 1st part of the manuscript. Note that South Africa is in fact a net importer of red meat. It is very difficult to quantify the value of the game meat industry, particularly as some individual animals fetch over US$1 million (https://businesstech.co.za/news/business/112621/this-buffalo-is-worth-more-than-what-top-south-african-ceos-earn-in-5-years/) – obviously these animals illnot be hunted for meat. However, their offspring that do not meet the required standards (be it hrons, sex, or pelt colour) are often harvested for meat – at a lower value. Also there is no data base in South Africa indicating the number of wild ungulates in the country – in fact data is only known for the formal commercial agriculture  sectors as pertaining to the guesstimate of the livestock numbers, this does not include numbers for livestock farmed in the informal sector.

Comment 3

  1. No summary on findings based on previous studies mentioned in Line 54-62ate.

Response to comment 3

Thank you, findings of the previous studies are summarised and captured in the manuscript especially the last part of the introduction. The findings have made it necessary to develop this plan for meat production. To also be precise and clearer, we have summed up the finding and made references to the different conclusions of published work on these issues. 

Comment 4

  1. No objectives stated in the manuscript

Response to comment 4

The rewritten manuscript has included the objective in the last paragraph of the second page and incorporated this information to explain the need for such a plan especially in game meat processes of slaughter and meat production.

Comments 5

  1. Define role-players (line 65). What is the difference between stakeholders?

Response to comment 5

The stakeholders and role players could be described as people or organisations that have a meaningful role in processes linked with game meat production. These organisations and their role have been captured in the discussions and reference to the suggested model for game meat production in the manuscript. 

Comment 6

  1. What is FSMS (Line 87)?

Thank you, FSMS is Food Safety Management System. This has been clarified and all acronyms have been written in full the 1st time they appear in the manuscript.   

294 to 306: again, linking these activities with the concept of "animal welfare" simply sound as absurdity.

Response to comment 6

Thank you, this is similar to the comments made by other reviewers and reference to welfare of the animal has been removed across the manuscript.

Comment 7

Define small game killings

Response to comment 7

The description of small game and examples thereof has been included in the manuscript – ‘small game’ is also defined in the Meat Safety Act (https://www.gov.za/sites/default/files/gcis_document/201409/a40-000.pdf)

Veterinary Procedure (VPN) for registration of hunters and game farmer/s

(http://www.old.dalrrd.gov.za/vetweb/VPN%20&%20SOP/008-VPN%2008%20Standard%20for%20the%20registration%20of%20hunters%20for%20harvesting%20wild%20game%20intended%20for%20export%20of%20game%20meat..pdf)

Comment 8

  1. Methodologies are not clear and comprehensive. What kind of review do you opt?

Response to comment 8

This paper follows a narrative review approach. This review stems on from 6 PhD papers that have been published: 2 papers looked at each possible hazard (Microbiological – Total Plate Count, Coliform, Ecoli and Salmonella); Chemical hazards (Lead, Cadmium, Arsenic and Mercury) and Physical hazards (Ammunition particles dispersion and bones splinters). This 7th paper combines all the efforts and thus identifies the key stakeholders in the industry and develops a basic plan for hygiene control of small game slaughter. A reviewer form one of the earlier papers suggested that such an encompassing paper describing this plan would be welcome.

Comment 9

  1. Based on theoretical model & basic meat safety plan, what are your conclusions?

Response to comment 9

The general fragmentation of hygiene controls, activities and actions by different role players has led to the proposal that game meat animal slaughter stakeholders must come together in a forum and discuss issues linked with these processes. The authors used published articles to develop a basic hygiene plan that could be discussed in this forum and implemented as needed. This information is included in the manuscript.

Comment 10

  1. No references for the theoretical model and basic meat safety plan under results

Response to comment 10

The concept of the model is developed and packaged by the authors, this is similar to the basic plan.

Comment 11

  1. If you are drafting the hazard identification planning & intervention strategies for small game killing, what are you proposing?

Response to comment 11

Thank you, the proposal being carried out is that game slaughter or stakeholders as identified, must follow the guidelines and identify the needed interventions to enable the production of safer meat. In this forum, training, refreshers and updates on regulations will be done, this forum could be used as information sharing platform for personal development. The basic plan will then be implemented during the actual killing or slaughter of small game (especially the control column)

Comment 12

  1. The authors do not state the importance of hazard identification in small game killing in regards to meat safety, etc. Provide justifications and research findings to support your problem state

Response to comment 12

Thank you, in the manuscript the risks of consuming hazards are clarified. The authors have also included not only hazards identification but also possible controls implementable during the slaughter of these animals

Reviewer 4 Report

The study presents interesting data related to “ Basic hazards control plan of small wild ungulates slaughtered for meat production purposes”.

However, the work in its actual form, has major deficiencies that are described in the manuscript. Furthermore, we strongly recommend an English revision by a native person. For these reasons, in my personnel opinion, this submission should be major revision.

Author Response

Response to Reviewer 4 Comments

Comment 1

The study presents interesting data related to “ Basic hazards control plan of small wild ungulates slaughtered for meat production purposes”.

Response to comment 1

Thank you.

Comment 2

However, the work in its actual form, has major deficiencies that are described in the manuscript. Furthermore, we strongly recommend an English revision by a native person. For these reasons, in my personnel opinion, this submission should be major revision.

Response to comment 2

Thank you for the guidance. All suggestions in the edited file you have prepared were included. Your comments in the document have led to the repackaging of this manuscript it has improved. Thank you. The paper has been read by a native English speaker.

Round 2

Reviewer 2 Report

I believe that this new version of the article proposal is more straightforward with purposes, with respect to previous version.

The article proposal now, in its new frame (food-safety), is really interesting.

Congratulations

Author Response

Response to Reviewer 2 (2) Comments

Comment 1

I believe that this new version of the article proposal is more straightforward with purposes, with respect to previous version.

Response to comment

Thank you, thank you for the guidance and thumbs up. We trust that this manuscript will add much needed knowledge.

Comment 2

The article proposal now, in its new frame (food-safety), is really interesting.

Congratulations

Response to comments 2

Much appreciated and we thank you for the insightful guidance and comments. They have added value to the quality of the manuscript.